# Paroxysmal Atrial Fibrillation in Horses: Pathophysiology, Diagnostics and Clinical Aspects

**DOI:** 10.3390/ani12060698

**Published:** 2022-03-10

**Authors:** Sofie Troest Kjeldsen, Sarah Dalgas Nissen, Rikke Buhl, Charlotte Hopster-Iversen

**Affiliations:** 1Department of Veterinary Clinical Sciences, Faculty of Health and Medical Sciences, University of Copenhagen, Agrovej 8, DK-2630 Taastrup, Denmark; rib@sund.ku.dk (R.B.); charlotte.hopster-iversen@sund.ku.dk (C.H.-I.); 2Laboratory of Cardiac Physiology, Department of Biomedical Sciences, Faculty of Health and Medical Sciences, University of Copenhagen, Blegdamsvej 3B, DK-2200 Copenhagen, Denmark; sarahnissen@sund.ku.dk

**Keywords:** paroxysmal atrial fibrillation, arrhythmia, equine cardiology, electrocardiogram, poor performance, long-term monitoring device, implantable loop recorder

## Abstract

**Simple Summary:**

Atrial fibrillation is the most common pathological cardiac arrhythmia affecting performance in horses. The sporadic form of atrial fibrillation, termed paroxysmal atrial fibrillation, spontaneously cardioverts to normal sinus rhythm usually within 7 days. The condition may go undetected, as episodes can occur intermittently at different frequencies and vary in duration from minutes to hours. However, paroxysmal atrial fibrillation may still result in poor performance, especially in racehorses, resulting in possible negative consequences for the horses and their owners. The epidemiology and pathophysiology of the disease are not well described. We investigate the current knowledge and present possible risk factors that may predispose horses to paroxysmal atrial fibrillation. Early diagnosis is crucial, which is why current and future diagnostic modalities are discussed.

**Abstract:**

Atrial fibrillation (AF) is the most common arrhythmia in horses causing poor performance. As in humans, the condition can be intermittent in nature, known as paroxysmal atrial fibrillation (pAF). This review covers the literature relating to pAF in horses and includes references to the human literature to compare pathophysiology, clinical presentation, diagnostic tools and treatment. The arrhythmia is diagnosed by auscultation and electrocardiography (ECG), and clinical signs can vary from sudden loss of racing performance to reduced fitness or no signs at all. If left untreated, pAF may promote electrical, functional and structural remodeling of the myocardium, thus creating a substrate that is able to maintain the arrhythmia, which over time may progress into permanent AF. Long-term ECG monitoring is essential for diagnosing the condition and fully understanding the duration and frequency of pAF episodes. The potential to adapt human cardiac monitoring systems and computational ECG analysis is therefore of interest and may benefit future diagnostic tools in equine medicine.

## 1. Introduction

An unexpected and abrupt decrease in performance in horses without any previous clinical signs may be related to the sudden onset of cardiac arrhythmias [1]. Atrial fibrillation (AF) is the most common pathological arrhythmia in horses with poor performance. However, as studies investigating the presence of AF employ different methodologies and study different populations, the reported incidence and prevalence vary greatly from an estimated prevalence of 0.29% [2] and an estimated incidence of 4.9% in Thoroughbreds with reduced racing performance [3] to an overall prevalence of AF in Standardbreds of 0.11% [4]. In hospitalized mixed-breed populations, the prevalence of AF has been reported at around 2.3–6.2% [5,6]. The different nature of the studies limits their translatability, and a sufficient estimation of the true burden of AF is still required. The sporadic form of AF is termed paroxysmal AF (pAF) and is characterized by intermittent episodes of AF that spontaneously cardiovert to normal sinus rhythm (SR) [7,8]. In recent decades, numerous studies have investigated the mechanisms of AF in horses along with different treatment options and the effect on performance [9,10,11]. However, less is known about pAF, likely due to its sporadic nature and lack of clinical signs in non-racing horses. The arrhythmia can easily be overlooked if the horse is not examined during the short window of time when the arrhythmia is present, and it is therefore likely to be underdiagnosed [7,8,12].

Poor performance is a phrase commonly used to describe horses that do not meet the expectations their owners. However, veterinarians must consider breed, age, discipline and training status of the horse when evaluating the level of performance. In relation to pAF, poor performance has been used to describe even intermittent loss of physical capacity in racehorses, whereas for riding horses it describes a loss of energy and inability to perform the requested skills.

It is easier to recognize pAF in humans, as many patients experience symptoms such as palpitations and increased heart rate (HR), but also because home monitoring with either long-term electrocardiograms (ECGs) or self-monitoring wearables helps diagnose sporadic episodes of pAF [13,14]. In equine medicine, the challenge is early detection and recognizing horses at risk [12]. More knowledge about pAF is needed, which is why we dissect here the current literature on pAF in horses and discuss possible future diagnostic modalities in equine cardiology.

## 2. Definition of Paroxysmal Atrial Fibrillation

AF is a supraventricular arrhythmia characterized by an irregularly irregular heart rhythm [7,8,15], which may occur either secondary to an underlying disease (e.g., myocardial disease, mitral regurgitation or pulmonary disease) [16,17] or without any concomitant disease. The terms “idiopathic AF” or “lone AF” have previously been used to emphasize the absence of underlying disease, yet this should be avoided as even subclinical pathophysiological changes in the atria are considered to be “underlying” causes [18]. In human medicine, AF has been defined as “paroxysmal” if it spontaneously cardioverts within 7 days without treatment, “persistent” if it lasts longer but ultimately responds to treatment and “permanent” where there is no response to treatment [19,20,21,22]. The human classification system has been adapted for equine medicine [23], but the exact definition of pAF is still debated. According to Holmes, pAF is thought to self-terminate within 24 h, and in one study horses that spontaneously cardioverted after 24–45 h were therefore diagnosed with persistent AF [8]. Reef et al. defined pAF as self-terminating within 24–48 h [24], Nath et al. within 72 h [3] and van Loon within 5 days [1]. An exact definition is necessary in order to compare future studies but also to give clinicians information about prognosis and possible treatment. We recommend integrating the human classification and defining pAF as AF that self-terminates within 7 days. In this review, “AF” includes both persistent and permanent AF.

## 3. Pathophysiology

The complex pathophysiological mechanisms of AF have primarily been investigated in human clinical cases, in *in vivo* animal models (including rodents, dogs, goats, pigs and horses) [10,25,26] and *in vitro* experiments at cellular and molecular levels [27]. In this review, we focus on the early stages of AF and the progression of pAF to persistent AF (Figure 1). For a detailed description of AF pathophysiology, readers are referred to a recently published review of AF in horses [28].

Episodes of AF are characterized by rapid and chaotic electrical activity in the atria causing reduced atrial function. The arrhythmia arises when there is an abnormal atrial depolarization together with the presence of disturbed conduction throughout the atria. This combination is believed to facilitate the onset of AF and several mechanisms are thought to be involved in the process, as briefly explained in the following section.

### 3.1. Triggers

Focal spontaneous firing (ectopic activity) may trigger AF. At high rates, ectopic firing alone can drive AF, while a single spontaneous firing requires a vulnerable substrate in order to induce AF. Triggering activity may have different causes, including enhanced cellular automaticity and abnormal automaticity from ectopic regions caused by Ca^2+^ handling abnormalities [29,30]. These triggers can be either promoted or suppressed by altered neurological stimulation (sympathetic or parasympathetic) [31], and susceptibility to AF induction can be enhanced by stretch of the atrial myocardium [30].

Ectopic firing can originate from foci anywhere in the atria and may result in a premature atrial complex (PAC) when it occurs inappropriately early. Frequent PACs—and several within a short time interval—are believed to be potential triggers for AF in horses [7,24]. Some clinical reports on pAF in horses observed frequent PACs, either before onset—representing a potential triggering cause [32]—or after an episode of pAF [7]. However, it is uncertain whether a high burden of PACs increases the risk of developing AF [33]. Atrial myocytes with altered molecular properties can function as ectopic firing foci. Cells with membrane potentials similar to pacemaker cells have been identified in horses, and these may have originated from the same embryonic cells as myocardial conduction tissue [34], commonly located near the pulmonary veins (PV) in humans [35]. Pulmonary vein firing is a common trigger for pAF in humans [35,36], while its role in horses is currently unknown. Spontaneous atrial firing originating in the area of the pulmonary veins was recently reported in a horse undergoing atrial mapping [37], and it is likely that PV activity could play a role in the initiation of AF in horses.

Molecular changes in the Ca^2+^ handling apparatus have been identified in human pAF patients, where ryanodine receptor 2 (RyR2) dysregulation is present, which may cause Ca^2+^ leakage from the sarcoplasmic reticulum leading to delayed after-depolarizations, thus triggering AF [38]. In addition to Ca^2+^-dependent triggered activity, cellular coupling between atrial myocytes and activated myofibroblasts investigated by computational analysis has been proposed to cause ectopic activity in humans with pAF [39,40]. It is unknown whether the same mechanisms exist in horses. However, as myofibroblasts are relatively common in equine atria, these cells are indeed of great interest and should be investigated more closely [41].

### 3.2. Substrate

As previously mentioned, a single spontaneous firing requires a vulnerable substrate in order to elicit an episode of AF. The substrate consists of myocardial tissue with an altered composition or function due to remodeling, which promotes the maintenance of arrhythmias. If the triggering activity is not supported by a proper substrate, the fibrillating wave will terminate, which could be the case with pAF. Experimental animal studies have shown that pAF induces atrial remodeling, which could potentially promote progression from pAF to permanent AF [25,42].

### 3.3. Atrial Remodeling

AF causes electrical, structural and functional remodeling of the atria. This forms the basis of a vicious cycle as the AF-induced remodeling acts as a substrate that allows AF to become self-sustained [5,10].

The electrical remodeling can be assessed by measuring the atrial fibrillatory rate (AFR) and the correlating atrial fibrillation cycle length (AFCL), which shortens as the AFR increases. Studies in horses with AF measured the AFR of surface ECGs and found that horses with short-term AF (<24 h) had a significantly lower AFR compared to individuals with long-term AF (55 days) [10,43]. In humans, a low AFR has been shown to be predictive of self-termination in patients with a recent onset of AF [44]. The atrial effective refractory period (aERP) is an important electrophysiological parameter correlated with the AFCL. Both the aERP and AFCL shorten with increased AF duration, suggesting electrical remodeling, which in turn increases the AFR [45]. An equine model of short-term induced AF demonstrated how the aERP was markedly reduced after only 12 h of AF, yet when SR was restored after 7 days, the aERP returned to normal within only 24 h [46]. A similar electrophysiological response was found in goat models [25,47]. While electrical remodeling is likely to be relevant to the process, the fast normalization after SR restoration suggests that electrical remodeling alone cannot explain the progression from pAF to AF, and other factors, such as early structural changes, may also influence this development.

Functional remodeling can be measured by decreased atrial systolic function and altered atrial dimensions. After cardioversion (spontaneous or induced), a transient mechanical dysfunction of the atria (also known as atrial stunning) is present, which might be caused by a reduction of the L-type Ca^2+^ current [47,48]. One study identified a reduction in atrial contractile function when inducing short-term AF (7 days) that mimicked pAF in horses [46]. Similarly, horses with AF undergoing induced cardioversion have decreased atrial contractility initially after cardioversion, but the atrial remodeling was reversed at a long-term follow up after 7 weeks [49]. However, the AF duration and the time taken for the remodeling to be reversed are likely to be correlated [10,49,50]. An association between the mechanisms of functional and electrical remodeling is suggested due to their synchronicity during the reversed remodeling in SR [47].

Structural changes, such as cellular hypertrophy, cellular dedifferentiation to a precursor and interstitial fibrosis have been reported in horses with persistent AF [5,10,51,52]. Studies investigating structural changes in response to AF found an increased number of atrial fibroblasts after 6 weeks of induced AF [41] and increased atrial fibrosis after 2 months of induced AF [10].Two major inflammatory drivers involved in fibrosis formation are angiotensin II and transforming growth factor β1 (TGF-β1). These factors are heavily involved in fibrogenesis, including the transformation of fibroblasts into collagen-producing myofibroblasts, and increased tissue levels have been found in the right atrium of patients with pAF. The presence of fibrosis undoubtedly leads to conduction abnormalities that enhance electrical re-entry [53,54,55,56,57], and early onset of these mechanisms should also be expected in equine pAF patients.

## 4. Epidemiology

The first report of equine pAF was in the 1950s [58,59]. The condition is primarily diagnosed at post-race check-ups of slow or non-finishing racehorses [2,3,7,60] or while investigating reduced performance [12,61]. The condition can be caused by underlying diseases affecting cardiac function, such as severe airway disease, where the right atrium will enlarge in response to reduced right ventricular function [16]. Large-scale studies have identified pAF in slow-finishing racehorses [2,3]. One study found a prevalence of 0.29% in Thoroughbreds; however, they only included slow-finishing horses and excluded non-finishers, causing 28 pAF episodes to be excluded from the analyses [2]. This example emphasizes the importance of examining both slow-finishing horses and non-finishers after racing and including all diagnosed cases in order to report the true prevalence or incidence of pAF. An overview of the reported pAF cases is presented in Table 1.

### 4.1. Breed and Heritability

Most cases of pAF have been reported in Thoroughbreds [2,3] and Standardbreds [4,12], but it has also been reported in other breeds [16] (Table 1). AF is more common in larger horses compared to ponies [6]. A retrospective study from Canada reported that Standardbreds were more commonly affected by AF than other breeds [64]. It was later discovered that AF is a moderately heritable arrhythmia in Standardbreds, and frequently used breeding stallions might have helped increase the incidence, though the mode of inheritance is still unknown [65,66]. A recent study investigating heritability in Thoroughbreds from Hong Kong and Australia found the heritability of AF to be low [67]. In humans, the heredity of AF is only linked to the underlying cardiac condition and not AF itself [20].

### 4.2. Sex

Most cases of pAF occur in male horses [2,3,7,8,12,61], but, as highlighted by Nath et al. and Reef et al., this reflects the equine population on the race course, and no significant difference in the frequency of pAF between sexes has been found [2]. Likewise, no difference between sexes has been found in human studies, except that females with AF are at greater risk of thromboembolic stroke [20,68].

### 4.3. Age

Similar to humans, the risk of AF in horses increases with age. The ageing human patient develops atrial remodeling preceding AF and, as explained earlier, this includes fibrosis, altered connexin expression and Ca^2+^ dysregulation [69]. In addition, progression from pAF to AF is more frequent in older human patients [70]. No difference was found in the prevalence of AF when comparing horses that were 4 years old with horses that were 5 years or older in a population of hospitalized horses [17]. Another study found the risk of AF increased in racehorses that were 4 years or older [2] and this was supported by Slack et al., who found age to be a risk factor for AF in Standardbreds [4]. The latter studies both examined horses with no underlying conditions, whereas Deem and Fregin examined hospitalized patients, several of which had underlying conditions, thus resulting in a high number of variables that might affect the risk calculations. Although no studies have demonstrated any association between equine pAF and age, it appears from the literature that younger horses are mainly affected (Table 1). However, racehorses represent a young population, which increases the risk of selection bias. Three cases of pAF were reported in neonate foals [63], but the triggering mechanisms are thought to be different compared to adult horses, and whether or not the incidence of equine pAF increases with age remains unknown.

### 4.4. Predisposing Factors

Larger animals (horses, cattle and giant breed dogs) are thought to be more prone to developing pAF in the absence of other diseases compared to smaller animals (other dog breeds, pigs and goats) [71,72,73]. Larger animals have a larger atrial mass, allowing more re-entry circuits to coexist [74]. Similarly, larger atrial size and tall stature increase the risk of AF in humans [75,76]. Furthermore, a large atrial size has been shown to increase the risk of progression from pAF to permanent AF in humans [70]. This might explain why large horses without structural heart conditions are more prone to developing AF compared to ponies. To the authors’ knowledge, there are no reports on naturally occurring pAF in ponies, only experimental models of induced AF [77,78]. Although a relatively large atrial size is a possible risk factor for developing AF in many large horses or giant breed dogs, the majority of these animals never develop AF. This suggests that factors other than atrial size must also play a role.

Valvular regurgitations are common in horses and potentially predispose for AF, yet mild valvular regurgitations are not associated with AF [61]. However, moderate mitral regurgitation is a known risk factor as it will lead to atrial volume overload and therefore an increased atrial pressure and myocardial stretch, which can trigger AF [79].

There are several known risk factors for human AF, including age, hypertension, valve disease, sleep apnea and obstructive lung diseases [75], whereas only age or valvular disease seem to be of relevance in horses.

### 4.5. Autonomic Nervous System

The autonomic nervous system consists of sympathetic and parasympathetic compartments, both interacting with cardiac electrophysiology. Altered autonomic tone affects the electrophysiological properties of the atria and plays a role in the onset of AF [27,80,81]. A circadian variation of pAF has been reported in both humans and dogs [82]. In humans, an increased episode duration occurs at nighttime, with an increased number of episodes during the early morning and evenings [83,84]. This suggests that AF onset is more common when the parasympathetic tone dominates. Interestingly, Buhl et al. reported similar findings in horses, where the majority of pAF episodes diagnosed with implantable loop recorders (ILR) also occurred while the horses were resting and were thus dominated by the parasympathetic tone [12]. This contradicts previous reports on pAF in racehorses (Table 1), in which the onset of pAF occurred during strenuous exercise, but different triggering mechanisms may explain this [3,7,8]. It is worth noting that different diagnostic modalities were used in the different studies, as discussed later. However, many transient arrhythmias appear in the transition from sympathetic to parasympathetic dominance, e.g., after racing [85,86].

## 5. Clinical Presentation

Immediate clinical examination of horses with AF will reveal the following: irregular heart rhythm, delayed recovery of the HR and respiratory rate, and variation in heart sound intensity [7,15,60,62]. In rare cases, horses can show distress [8]. The majority of horses with AF have a normal ventricular rate at rest, while some have an increased resting heart rate [87]. The conduction of atrial signals through the atrioventricular node are believed to be affected by several factors, including the autonomic tone, concealed conduction and the AFR [9,59,88].

In racehorses, AF is commonly characterized by an instant loss of speed that causes the horse to finish markedly behind the winner [3,7,8,15]. This is caused by the sudden lack of peripheral oxygenation because AF impairs ventricular filling and affects the cardiac output [8]. If the onset of AF begins at the finish line or during deceleration, it will not affect the horse’s racing ability [4,32,62]. In fact, 3 out of 13 horses won their race despite being diagnosed with post-race AF [62]. In pleasure horses, clinical signs are thought to be more subtle or they may even be asymptomatic due to less stress on the cardiovascular system. The sporadic nature of pAF may explain why some horses present with intermittent episodes of reduced performance [12].

### 5.1. Foals

Three cases of pAF were diagnosed among 20 neonate foals at birth (in 12 of these, fetal ECG recordings confirmed SR 1 week before parturition). All three foals were markedly depressed after the onset of breathing and an ECG confirmed AF. In two out of the three foals, pAF lasted for 170 and 200 min, and the clinical signs included pale and cyanotic mucous membranes, tachypnea and tachycardia. In one foal, the episode lasted only min, but none of the foals were able to stand before they were in SR, after which all clinical signs resolved. The triggering factor was likely the onset of breathing, which caused increased pulmonary blood flow and increased left atrial pressure, resulting in atrial stretch [63].

### 5.2. Underlying Disease

There were no underlying predisposing diseases in the majority of pAF cases reported (Table 1). Likewise, Reef reported that 56.7% of horses with AF had no other cardiac diseases [61]. However, myocardial disease was suspected to be the cause of pAF [15] and AF [5] in previous reports. The vast majority of human patients with pAF progressing to AF have underlying diseases, such as hypertension, valvular disease, heart failure or chronic obstructive pulmonary disease [70]. Pulmonary hypertension and progression to cor pulmonale causes increased right atrial pressure and volume overload, which promotes atrial remodeling, acting as a substrate for AF, and has been described in both horses and humans [75,89]. One horse with recurrent airway obstruction developed cor pulmonale and pAF. Echocardiography revealed a markedly enlarged right atrium and ventricle with valvular regurgitations (tricuspid, mitral and pulmonary) and the pulmonary artery pressure was acutely increased. The affected horse developed pAF that lasted for 2 days and self-terminated when the underlying disease was treated [16].

## 6. Diagnostic Modalities

During auscultation, AF is recognized as an irregularly irregular heart rhythm without the fourth heart sound, while the ECG will show irregular RR intervals, an absence of P-waves, presence of fibrillation waves (f-waves) and predominantly normal QRS complexes [1]. Episodes of pAF vary in both duration and frequency (Table 1) [12]. The duration of a single episode was reported to be as short as 2 min when detected by an ILR [12] and up to 48 h [16]. The AF burden can be defined as the proportion of time an individual is in AF during a given monitoring period [90]. The total AF burden of an individual can be used to measure the possible progression of AF. The true episode duration and frequency might be underestimated if not diagnosed with an ILR because the time of onset can otherwise only be estimated based on history and clinical findings [61,91].

### 6.1. Long-Term Cardiac Monitoring Devices for Veterinary Use

Holter ECG monitoring is currently considered the gold standard for diagnosing arrhythmias [92]. In order to diagnose a sporadic arrhythmia like pAF, long-term cardiac monitoring is essential [12]. Cardiac monitoring devices detect arrhythmia by measuring heart rate variability (HRV), which describes the beat-to-beat variation represented by RR intervals. In a resting horse, HRV is affected by 2AVB and sinus arrhythmia, resulting in an increased HRV [85,93]. A HRV that exceeds a certain percentage indicates the presence of arrhythmia, and this has been used to distinguish AF from SR [94]. One study examined horses undergoing a standardized treadmill exercise test to investigate the percentage of RR shortening that would be predictive of an arrhythmic event. The Youden’s index cut-off of the RR variation was found to be 5.7% in trot, 11% in canter and 6% during the recovery phase. However, the type of arrhythmia was not specified. 

One study used an ambulatory ECG pad (Carnation Ambulatory Monitor™, Bardy Diagnostics Inc., Seattle, WA, USA) developed for humans to record a 7-day ECG in horses with suspected arrhythmias [95]. The pad was placed externally on the horse in different positions, where the ventral midline caudal to the xiphisternum has fewer artefacts. However, when positioned on the left side of the thorax, atrial signals (p-waves) were more visible. In general, lack of skin contact and poor-quality ECG during exercise are considered to be limitations of this product. Another study tested whether ILR (Reveal LINQ™, Medtronic Denmark A/S, DK-2300 Copenhagen S, Denmark) could detect AF in horses [91]. The ILR is implanted subcutaneously in the 6th intercostal space on the left side of the thorax. It continuously monitors for possible AF episodes by measuring RR interval differences in a Lorenz plot and monitors for the presence of a P-wave. The ILR saves the recorded ECG segment and registers the time of onset and duration of the episode. This method was used to detect pAF in four out of 12 Standardbred trotters with poor performance over a monitoring period of 6–28 months [12]. The results of this study emphasize the importance of long-term cardiac monitoring in detecting pAF but highlight the potential increase in time until a diagnosis. Common to both the ECG pad and the ILR is that an experienced clinician must analyze the recorded ECG to avoid false-positive detection of pAF due to artefacts or other arrhythmias. One major limitation of both products is their inability to detect pAF during exercise due to artefacts [12,95]. Larger studies, including exercise testing, would contribute more information about these potential future diagnostic methods. Other limitations include the cost of the products and their availability in general veterinary practice.

### 6.2. Cardiac Home Monitoring Devices for Horses

Several pulse girths and fitness trackers are commercially available for horse owners, promoted to monitor the fitness level and cardiac health of their horses. These devices are able to measure HR and HRV with incorporated software [96,97,98,99]. When comparing HRV parameters from the commercial products with gold-standard ECGs, manual correction of the data from commercial products is essential, especially for detecting arrhythmias [96,98]. As with all externally attached devices, increased movement resulted in an increase in artefacts, and these products are only able to collect good quality data when the horse is at rest [96,97,99]. Recently, a smartphone ECG device was used to detect pAF in Thoroughbreds after racing, as the algorithm can distinguish SR from AF and can therefore be used as a screening tool [100]. Although some devices have been shown to distinguish SR from AF [100,101], they cannot replace a veterinary examination [98].

### 6.3. Continuous Cardiac Monitoring in Humans

The AF burden can be measured either internally via insertable cardiac monitors (e.g., implantable loop recorders) or externally by Holter ECG or wearables (such as bracelets/smartwatches). Human patients with AF sometimes feel palpitations when in AF, but by using long-term continuous monitoring devices, the true AF burden can be registered, including during the night; this can help elucidate any event that may have triggered the AF. Newer wearables, such as smartwatches, can be used to self-monitor episodes of AF. Wrist-worn wearables use photoplethysmography, where a photosensor detects blood flow through arterioles using green LED light deflection [102]. By measuring HR, they can calculate HRV and thereby detect possible arrhythmias, which may be useful for AF detection and self-monitoring. A review comparing a number of studies that tested devices with photoplethysmography highlighted that these were mainly carried out in ambulatory facilities, only measured at short time intervals, and did not account for motion artefacts [102]. However, self-monitoring AF can have negative consequences if it triggers health anxiety through a fear of stroke or worsening of the disease [103]. The technology behind these devices is developing rapidly but cannot yet replace insertable cardiac monitors. Devices that use photoplethysmography technology might be difficult to use in horses, as pigmented skin and hair will affect the data quality [104].

### 6.4. Computational Analysis of ECGs

Computational analysis of ECGs of apparently normal SR have recently been tested to identify horses that are more likely to develop pAF [105,106]. The artefact-free ECG string of 60 s was converted to binary sequences using different filters: one filter registers threshold crossing; another filter registers beat detection, where the value at the R-peak has a value of 1 and the remaining values are 0; the third uses feature detection, where several specific ECG features, such as onset-Q, Q-peak, R-peak, R-end, T-peak and T-end, were converted to a binary sequence. The computational complexity in ECGs was significantly higher in horses that developed pAF. Only short ECG sequences of 60 s were tested and only at HR 25–60 per minute [105]. The same group also investigated the possibility of using machine learning to analyze the ECG restitution, in order to reveal a proarrhytmic substrate in horses with pAF. The model measures the basic cycle length (RR interval), action potential duration (QT interval) and diastolic interval (TQ interval) to quantify the restitution of every heartbeat. The machine learning algorithm then compared the distribution of data from healthy controls with pAF horses [106]. These novel methods of assessing ECGs from healthy horses and the possible early diagnosis of pAF could lead to close patient monitoring and thus early treatment, yet a strong algorithm is required, which is why further studies are needed. In future diagnostics, when artificial intelligence becomes more and more powerful, computational ECG analysis will be an important dimension and may play a role in future neural networks for diagnosing pAF.

## 7. Treatment

### 7.1. Human Treatment Strategies

In humans, AF is associated with a reduced quality of life and increased morbidity and mortality due to several factors, including atrioventricular dyssynchrony, progressive cardiac dysfunctions and altered hemodynamics [20,107]. Unlike horses, humans with pAF are at increased risk of stroke due to reduced atrial systole and increased atrial stasis [108], while pAF also initiates an early state of hypercoagulability [109]. A study investigating hypercoagulability in horses with AF could not support findings from human studies [110]. The differences in atrial clot formation in humans versus horses warrants further investigation.

The first step in treating human patients with AF is to avoid thromboembolic strokes by assessing the patient’s stroke risk, and in patients with increased risk oral anticoagulants (OAC) should be administered [20]. Scoring systems (CHA_2_DC_2_-VASc and CHAD_2_) that take into account demographics and health history are recommended for risk stratification [111,112]. Only assessing the AF type and burden when evaluating the risk of stroke may result in several important risk factors being overlooked [90]. However, when only comparing low-risk patients, clinical records show that patients with pAF with or without OAC have a lower risk of thromboembolic strokes and a lower risk of mortality compared to patients with persistent AF [90].

Anti-arrhythmic drugs (e.g., flecainide or amiodarone) can be administered intravenously to restore SR in the patient. If the drug is considered safe for the patient, a self-administered “pill-in-the-pocket” strategy may be beneficial in cases of pAF. However, if cardioversion is not achieved through the use of oral drugs or if AF becomes persistent, other strategies, including intravenous pharmacological cardioversion or electrical cardioversion, will be needed to restore SR [20,21]. Continuous medical treatment of horses with AF is limited, as these drugs are prohibited substances and only relevant to non-competing horses; further investigations of the long-term effect of these drugs are needed.

Radiofrequency catheter ablation (RCA) is a procedure used to scar the endocardium around the ectopic foci (e.g., pulmonary veins) to inhibit rapid firing signals in patients with AF. Before RCA, atrial mapping is used to localize the area of electrical activity of the re-entry waves. These methods have been described in horses [113,114,115] but more experience in mapping and RCA is needed before the techniques can be routinely used in equine patients. Clinical trials have concluded that human patients with pAF have a lower risk of recurrence when undergoing RCA compared to anti-arrhythmic therapy [116].

### 7.2. Veterinary Treatment Strategies

There are few treatment options for horses with pAF, but cardioversion is currently recommended if the episode lasts longer than 48 h [24,62]. Cardioversion can be achieved either pharmacologically with quinidine sulphate or by transvenous electrical cardioversion (TVEC), but these procedures are not within the scope of this review and are described in detail elsewhere [1,24,117,118]. Cardioversion of AF horses presenting with sudden reduced performance but without underlying diseases can return them to their previous level of performance [7,17,61].

## 8. Recurrence Risks

Second incidences of pAF in horses have been reported [3,7,12,61] and may occur anywhere from 34–1065 days after the first episode [3]. The overall recurrence rate following an episode of pAF was 25.1% in 4684 horses over a 10-year period. The recurrence rate in horses with persistent AF was 64% compared to only 23% in horses with pAF [3]. This may support the idea that AF duration is related to the risk of recurrence [61]. A multi-center study in AF horses undergoing cardioversion found a correlation between AFR and the risk of AF recurrence in horses undergoing TVEC [119]. AFR can be a measure of electrical remodeling and thus a predictor of recurrence risk in horses undergoing TVEC. The continuous remodeling of the atria serves as a substrate that maintains the disease. Horses with larger atria are predisposed to recurring AF [61]. Similarly, humans with larger atrial dimensions have an increased risk of AF recurrence [76] and an increased risk of progression from pAF to permanent AF [70].

The majority of reported cases with pAF (Table 1) all continued their careers without being affected by the transient episode and retired due to reasons not related to sporadic arrhythmia [7,8,15,62]. The number of pAF episodes did not affect the duration of their careers [3].

## 9. Conclusions and Future Research

Horses at risk of developing pAF are racehorses performing at a maximal level of exertion and sudden loss of speed is therefore the most common clinical sign. The estimated incidence in racehorses with poor performance is <5% when using short-term monitoring but 33% when using long-term monitoring devices. Large-scale longitudinal studies that include different breeds and equine disciplines of both apparently healthy horses and horses with reduced performance are needed to elucidate the true prevalence of pAF and its effect on performance. Long-term continuous cardiac monitoring is essential for diagnosing pAF, but suitable devices with high sensitivity during both rest and exercise are currently lacking. Better knowledge of the pathophysiology and mechanisms involved in the early remodeling process is needed to fully understand how pAF progresses into persistent AF. Insights into the pathophysiology of the remodeling process can also help to identify possible targets for future treatments. Future diagnostic techniques, including computational analysis of ECGs, could have considerable value in pAF screening and early diagnosis in horses.

## Figures and Tables

**Figure 1 animals-12-00698-f001:**
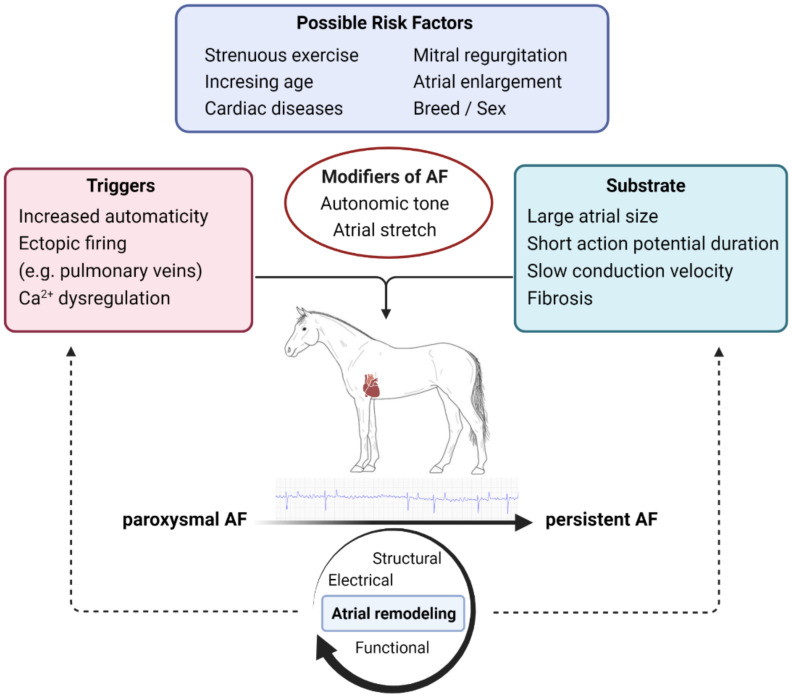
Possible risk factors and pathophysiology, including the triggers and substrate involved in the onset of paroxysmal atrial fibrillation (AF) and the progression to persistent AF in horses. Created with https://biorender.com/ (accessed on 2 March 2022).

**Table 1 animals-12-00698-t001:** Paroxysmal atrial fibrillation (pAF) in horses. The table provides an overview of previous studies reporting on pAF in horses, including specifics of the different study populations and disease duration. Horses with poor performance included: “Yes” if horses had poor performance on the day of examination, “No” if the horses included did not present with clinical signs. Number of horses with pAF only includes self-terminating episodes lasting less than 48 h. Breed: Thd = Thoroughbred, Std = Standardbred, Oth = other. AF = atrial fibrillation. Gender: M = male, F = female, with the predominant gender first. RAO = recurrent airway obstruction. Duration: d = day, h = hours, min = minutes. N/A = not available.

Year	Author	Study Design	Horses with Poor Performance Included	Horses with Underlying Disease Included	Sex	Breed	Examined (n)	pAF (n)	Age (Years)	Onset during Race (Yes/No)	Episode Duration
1952	Detweiler [59]	Case series	Yes	Yes	F	Std	5	1	8	No	10 h
1967	Glazier and Kavanagh [60]	Case report	No	No	F	Thd	1	1	2	No	1 d
1975	Amada et al. [7]	Case series	Yes	No	M, F	Thd	N/A	5	3–4	Yes	4 h, 20–23 h
1977	Rose and Davis [15]	Case report	Yes	No	M	Thd	1	1	4	Yes	12 h
1982	Deem and Fregin [17]	Retrospective study	Yes	Yes	M, F	Thd, Std, Oth	106	6	≥2	No	N/A
1986	Holmes et al. [8]	Case series	Yes	No	M, F	Thd	5	4	2–6	Yes	<24 h, 45 h
1987	Holmes [62]	Cross- sectional	Yes	No	N/A	Thd	19	11	Mean 5.6	Yes (n = 10)	<24 h
1988	Reef et al. [61]	Retrospective study	Yes	Yes	M, F	Std, Thd, Oth	67	3	Mean 6.6	N/A	N/A
1989	Machida et al. [63]	Case series	N/A	No	M	Thd	20	3	Neonates	N/A	<3 h
1999	Hiraga and Kubo [32]	Case series	No	No	M	Thd	2	2	2–5	No	4 min
2003	Ohmura et al. [2]	Case control study	Yes	No	M, F	Thd	8639	114	≥2	Yes	<24 h
2015	Hanka et al. [16]	Case report	Yes	RAO	M	Arabian	1	1	14	N/A	2 d
2015	Slack et al. [4]	Clinical study	No	No	M, F	Std	1816	3	Mean 4	Yes	4.5–12 h
2020	Buhl et al. [12]	Prospective study	Yes	No	M, F	Std	12	4	3–8	Yes (n = 2)	2–692 min
2021	Nath et al. [3]	Retrospective study	Yes	No	M	Thd	4684	230	Mean 5.1	Yes	N/A

## Data Availability

Not applicable.

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
