# Peer review of "Paroxysmal Atrial Fibrillation in Horses: Pathophysiology, Diagnostics and Clinical Aspects"

_animals, 2022, doi:10.3390/ani12060698_

Round 1
Reviewer 1 Report
I read with interest the manuscript ‘Paroxysmal Atrial Fibrillation in Horses: Pathophysiology, diagnostics and clinical aspects’. This manuscript is a review of the current literature.
I have some comments and areas that require further clarification.
In the abstract, the term symptoms is used several times. Horses do not have symptoms (this is a human phenomena) but rather clinical signs. Please take care to change this throughout the manuscript.
The opening paragraph of the introduction contains a summary of the known incidence and prevalence data on PAF in horses. I think the authors should more clearly establish that these are estimates, from a number of studies employing different methodologies, study samples and criteria for classification, therefore they are not directly translatable or comparable.
I’m not entirely comfortable with Figure 1. I would understand triggers to be the actual events that lead to the development of an arrhythmia (premature complexes arising from increased automaticity, triggered activity or early/late after depolarisations). However, I would think of atrial stretch (note the spelling error) and autonomic tone to be more ‘modifiers’ that increase or decrease the risk of a triggered event as opposed to the trigger themselves? I’m happy to be directed to the appropriate sources but reference 31 (mentioned below) does not elaborate on these enough to be convincing to me. Similarly, I would see large atrial size as the substrate and atrial enlargement (occurring as a pathological process) as a risk factor? Mitral regurgitation should be singular rather than pleural in this context as a horse presumably can only have one ‘mitral regurgitation’ at a time?
Line 94 onwards (section Triggers): As above – the authors cite reference 31 – which makes this statement about autonomic tone and atrial stretch being trigger of AF but does not include an appropriate reference to this statement. Please cite the appropriate references for these statements. I would suspect that it is the coupling interval (timing of the trigger) in combination with the other factors as described that creates the ‘perfect storm’ for AF to develop, or not. Certainly, there are far more atrial premature complexes that don’t cause AF than do. And it should be noted that there maybe many pathways to trigger AF – not just a single one.
Line 152: as it is difficult/impossible for us to measure contractility in the clinical setting, I think it would be more appropriate to say ‘atrial systolic function’ rather than contractility here?
Line 163: perhaps ‘precursor’ would be more appropriate than ‘preliminary’ here?
Line 166: fibroblast should be pleural I think.
Line 177: I don’t think you intend to say that PAF causing respiratory disease? But rather that severe respiratory disease leading to cor pulmonale (and therefore RA enlargement) might predispose to PAF?
Line 264: It’s not entirely accurate to say all horses with AF have a normal resting HR – this is true for the majority of horses, however there are occasional horses that develop AF and have a higher resting HR (in the absence of underlying heart disease). I would suggest modifying this statement to be less definite. In addition, it’s not just vagal tone that is responsible for the slower ventricular rate at rest, but also the process of concealed conduction (Gelzer, 2000 J Cardiovascu electrophysiol) and atrial fibrillatory rate.
Paragraph line 287: Is it the pulmonary hypertension it’s self or the cor pulmonale that subsequently occurs? – RA increase in pressure/volume and remodeling that is responsible for PAF? Please clarify.
Line 319: the cut off of 6% RR variation to detect arrhythmias is for arrhythmias at the trot or during recovery in the publication by Frick et al – during the canter the youden index cut off (best combined sp and sn) was 11% RR variation. Please be accurate when citing the literature.
Line 324: Similarly – the reference of Keen 2021 is misquoted – they decided on the ventral midline as the optimal position for placement of the ECG device, not the lateral thorax.
Line 400: Clot rather than cloth
Line 419: intravenous rather than intra venous
Line 423: e.g. rather than etc
Line 453: This might read better if you replaced ‘amount’ with ‘number’
Throughout the manuscript, there are some occasional grammatical/spelling errors – particularly the use of the word ‘why’ seems somewhat unusual. I would suggest that the authors carefully read through the manuscript to make sure that all the sentences make sense (e.g. lines 70, 350, 389, 457) and have the correct spelling and grammar.
Reviewer 2 Report
Kjeldsen et al presented a really interesting review about the impact of paroxysmal atrial fibrillation in horses, concretely in racing animals. This article is very well-written, well-documented and it provides a wide perspective of this pathology, including pathophysiology, diagnostics and clinical aspects. Moreover, another positive point is that a comparison to human or other species is included, increasing even more the relevance of this article. I think this article is very much ready to be published, however, I suggest the authors consult an English speaker for a thorough proof-reading of the whole manuscript.
A list of minor changes is included:
- Line 68: change to post-cardioversion
- Figure 1: change to “e.g”.
- Figure 1. Please include AF as abbreviation.
- Line 144: add a coma after however
- Line 147: “...remodeling is unequivocally relevant”
- Line 168: Please explain the meaning of TFGb1
- Line 175: change to “Since the 50´s”
- Table 1: I recommend to include table abbreviations at the end of the table
- Line 249: change to “occurs”
- Line 251: change to “AF onset is more...”.
- Line 272: change to “post-race”.
- Line 301: please check if it is correct “an irregularly irregular heart rhythm”
- Line 328: “it continuously monitors”
- Line 332: “the results of this study emphasize”
- Line 334: “an experienced clinican has to”
- Line 336: “abovementioned”
- Line 373: please remove the coma after SR
- Line 379: “several specific ECG features ... were converted”
- Line 433: “transvenous”
Reviewer 3 Report
This is a well-constructed and informative piece of work that will be of great value to those interested in the association between paroxysmal and persistent AF. The significance of paroxysmal AF in the horse has received insufficient attention to date and this summary is timely. My principal concern with the paper relates to the use of language, there being multiple examples of verb-subject disagreement and some relatively obtuse constructions which mostly obscure meaning but occasionally leave statements open to misinterpretation. The writing also seems to be quite patchy, suggesting different individuals have contributed to different pieces of the text and/or that the whole document has not been proofread by a single author with requisite skills in the use of English. As a result, while some points are made with great clarity and depth yet succinctness, others use obtuse wording, are imprecise, and often make literal statements suggesting a degree of simplicity in cause/effect relationships that is in fact either unclear or unproven.
These problems can be dealt with fairly quickly to ready the paper for publication. I provide specific examples below, but advise these should not be seen as comprehensive as I have only identified examples - the need for careful proofreading remains. In so doing, I have also raised points concerning content and interpretation that might usefully be addressed.
Line 14. Spontaneous cardioversion of AF can indeed occur anywhere from moments to days after onset and the underlying mechanisms remain unclear. There is also a marked species variation in the frequency, interval and circumstances with which this takes place, however. I recommend that this variation be given a little more space in the text and also that the novelty of the 7 day horizon, which is not universally adopted in horses, be emphasised.
Line 20." Current and future diagnostic modalities are discussed since early diagnosis is advisable"
Line 23. "In nature…
Line 29. "… Which over time may progress into…" The mistakes that can arise through direct species extrapolation (in this case, from humans to horses) can result in unforeseen outcomes and tunnel vision. This progression, though quite possible, has not been adequately demonstrated in the horse. Careful acknowledgement where extrapolation is employed should always be used.
Line 32. "… and may be future diagnostic tools…" Use of the definitive article ("the") is not necessary, implies singularity in the object to which reference is being made, and frequently appears inappropriately and without need in this paper.
Line 42. "… is estimated to be 0.11%…" All such numbers are estimates and should be described as such.
Line 65. "In human medicine, AF has been defined as paroxysmal if it spontaneously cardioverts within 7 days without treatment, persistent if it lasts longer but ultimately responds to treatment, and permanent if no treatment response can be achieved." Clearly, these definitions have to be retrospective.
Line 120. "It is unknown if the same mechanisms exist in horses. However, as myofibroblasts are relatively common in horse atria,…"
Line 124. This paragraph requires rewriting - at present it is extremely difficult to follow.
Line 132 - "… vicious cycle as the AF-induced remodelling acts as…"
Line 137.”… specific software" please explain what you mean - either name the software or describe what it is supposed to do.
Line 138. "In humans, a low AFR has been shown…"
Line 141. What is the meaning of "… and is essential in AF electrophysiology." ?
Line 146. The sentence starting on this line requires rewriting. The statements are too emphatic and go beyond what we currently know to be true, plus the structure of the sentence is difficult to follow.
Line 160."… reversed are likely correlated."
Line 161. "… of their synchronicity…"
Line 164. "… primitive state" or "… undifferentiated state…"
Line 179. Repeated use of the term "poor performing" or "poor performance" begs the question of what exactly this means. Certainly not the central issue in this paper, yet frequently used to connote a horse whose performance somehow does not rise to expectations. It is notable that performance issues in horses with pAF have typically been characterised as good races and bad races - inconsistency. If these terms are to be used there ought to be some definition of just what they mean. Alternatively, we might need to explicitly invoke a performance impact of pAF that is of even shorter duration than a single race, i.e., an arrhythmia that is intermittent on a moment-to-moment basis. Important because this paper attempts to characterise this arrhythmia and even describes clinical signs, yet fails to address this specific but potentially quite important point.
Line 181. "… which potentially excluded 28…"
Line 182. "… that the number of horses with pAF is larger than diagnosed…"
Line 183. Incorrect. This may be the case for the studies published, but is by no means an accurate statement in terms of describing what actually happens at the track - horses that fail to finish will most certainly be examined.
Line 184. "… cases is presented…"
Line 195. Be cautious in making this point - AF has been described as occurring spontaneously in mice - depends on the underlying mechanism. [Pan Z, Ai T, Chang et al, Am J Physiol Heart Circ Physiol. 2019, Rosenberg MA, Das S, et al, J Atr Fibrillation. 2012.]
Line 201. "… of AF to be low…"
Line 212. "… like progression from pAF to AF is more frequent…"
Line 217. "… risk factor for AF in Standardbreds" or "… risk factor for AF in standardbred trotters"
Line 221. "… pAF and age are associated…"
Line 227. Please rewrite this sentence, it contains two separate thoughts that need more careful separation for your meaning to be clear.
Line 231. "Also, a large atrial size has been shown to…"
Line 235. Again, the sentence starting on this line contains two separate thoughts that become confused when presented in conjunction. For example, it might better read as "Despite a relatively large atrial size being a possible risk factor for developing AF in many large horses and giant breed dogs, the majority of these animals never develop AF. This suggests that factors other than atrial size must also be present."
Line 238. Essentially the same issue. Valvular regurgitations do predispose but are not always associated with AF.
Line 246. Please rewrite this sentence.
Line 249. “…occurs…”
Line 251. "… suggests that AF onset is more common…"
Line 263. I wonder whether it might not be a little unwise to offer this description of the clinical presentation? Personally, I have never seen cyanosis in a horse with AF, at rest or after exercise, unless something else of great clinical significance was occurring simultaneously, and in which instance the AF itself was likely secondary and not primary. If the word cyanosis is to be used it might be best to offer some qualification.
Line 268. Suggest replacing this sentence with "This is caused by a sudden drop in cardiac output".
Line 275. I think you mean the horses, not the owners?
Line 296 "right atrium"
Line 313. I suggest rewriting this paragraph, which, compared with much of the rest of this review, presents a very simplistic perspective. Also, surely HRV has not replaced an ECG as the method of choice for detection and diagnosis of arrhythmia?
Line 328. "… for possible AF episodes…" - Surely characterisation of the arrhythmia as paroxysmal is a secondary step that will be based on the sum total of the information gathered. Also, use of a Lorenz plot represents a method for describing an arrhythmia and not a primary method of diagnosis. Reliable detection of P waves can rarely be achieved with such equipment and it is the absence of P waves and the presence of F waves that is employed in the diagnosis of AF, together with the lack of regularity.
Line 332. These statements might be accompanied by a recognition that this might be an inordinately long time in which to reach a diagnosis, a very real practical limitation.
Line 334. "… clinician has to…"
Line 392. I'm not sure use of this space to discuss human treatment strategies is necessary. I appreciate the guidance that such information can provide, but if this is to be presented it should be accompanied by some recognition of the potentially huge limitations that would be involved in the ongoing treatment of horses, particularly the justification for treatment in the face of the animal's use as an athlete.
Line 443. "This may support the idea…". Do we yet have sufficient information to make these statements - a little circumspection, perhaps?
Round 2
Reviewer 1 Report
Thank you for addressing my comments/concerns appropriately.